# Phosphorus Removal in VFCWs with Lightweight Aggregates Made of Fly Ash from Sewage-Sludge Thermal Treatment (FASSTT LWA)

Joanna Rodziewicz [1], Artur Mielcarek [1,*], Wojciech Janczukowicz [1], Andrzej Białowiec [2], Jorge Manuel Rodrigues Tavares [3,4], Krzysztof Jóźwiakowski [5] and Arthur Thornton [6]

1   Department of Environment Engineering, Faculty of Geoengineering, University of Warmia and Mazury in Olsztyn, Warszawska 117a, 10-719 Olsztyn, Poland
2   Department of Applied Bioeconomy, Wrocław University of Environmental and Life Sciences, 25th Norwida, 51-630 Wrocław, Poland
3   Department of Technologies and Applied Sciences, School of Agriculture, Polytechnic Institute of Beja, Rua Pedro Soares-Campus do IPBeja, 7800-295 Beja, Portugal
4   Fiber Materials and Environmental Technologies (FibEnTech-UBI), University of BeiraInterior, R. Marquês de D'Ávila e Bolama, 6201-001 Covilhã, Portugal
5   Department of Environmental Engineering and Geodesy, University of Life Sciences in Lublin, Leszczyńskiego 7, 20-069 Lublin, Poland
6   Atkins, Woodcote Grove, Ashley Road, Epsom KT18, UK
*   Correspondence: artur.mielcarek@uwm.edu.pl; Tel.: +48-89-523-38-46

**Abstract:** This study analyzed the effect of lightweight aggregates made of fly ash from sewage-sludge thermal treatment (FASSTT LWA) on the effectiveness of phosphorus removal from wastewater in vertical constructed wetlands (CWs), depending on FASSTT LWA content in the CW filling and hydraulic loading rate. It was performed over 13 weeks using 15 lysimeters prepared as double-layer systems. An upper layer was made of FASSTT LWA above the gravel layer with different thicknesses of FASSTT LWA (CW 0 cm: only gravel; CW 12 cm, CW 25 cm; CW 50 cm, and CW 100 cm: only FASSTT LWA). Each filling variant was repeated three times. Wastewater with a mean phosphorus concentration of 7.43 mgP/L was fed to the lysimeters once a day. The hydraulic loading rates tested were 3.0, 5.0, and 7.0 mm/d. Both the increased FASSTT LWA content in the CW filling and the decreasing hydraulic loading rate were found to boost the effectiveness of phosphorus removal in the treated wastewater. Constructed wetland filled in 100 % with FASSTT LWA ensured a reduction in phosphorus concentration below 2.0 mg P/L at all hydraulic loading rates tested.

**Keywords:** sewage-sludge thermal treatment; fly ash; lightweight aggregates; constructed wetlands; vertical flow; phosphorus removal





## 1. Introduction

Constructed wetland systems (CWSs) are deployed to treat multiple types of wastewater, including household sewage, industrial and agricultural wastewater, and polluted rainwater [1–4]. Today, they are one of the most important elements of household sewage treatment plants (HSTPs), especially in rural areas for single houses and small rural communities [5].

The effectiveness of operation, as well as the technological and technical reliability of constructed wetlands, depend primarily on the filling used [6]. At the same time, plants have a significant effect on the removal efficiency and mass removal rate of all pollutants, except phosphorous [7,8]. The filling determines the course and efficiency of filtration, sedimentation, adsorption, and ion exchange processes [9]. Low hydraulic conductivity can lead to the clogging of systems, resulting in reduced efficiency [10].

The filling also acts as a carrier on which the biofilm develops. Bio-treatment effectiveness depends to a large extent on the physical and chemical properties of the filling. The porosity of the material and the specific surface area provide microorganisms with better conditions for adhesion and development. Thus, the choice of filling type and its parameters (particle size distributions, pore spaces, degree of irregularity, and the coefficient of permeability) play a key role in ensuring the expected performance of a CWS-operated wastewater treatment plant [11].

So far, CW fillings have been made of the following materials: natural materials, such as sand, gravel, clay, vermiculite, calcite, marble, bentonite, limestone, shell, dolomite, shale, peat, maerl, zeolite, wollastonite [12–18]; and industrial by-products and solid wastes, such as fly ash, slag, blast furnace (BF) slag, steel slag, steel furnace slag, alum water treatment sludge, coal cinder, bauxite processing residue, iron ore and bauxite, oil palm shell, oyster shells, shell sands, crushed spent shells of shellfish, sugarcane bagasse, and opoka [19].

Other fillings have also been applied, e.g., activated carbon, calcium silicate hydrate, compost, bamboo charcoal, biochar, ceramsite, synthetic products, and lightweight aggregates (such as LECA, Filtralite, HelioFIR, Norlite, Phoslock, and Polonite) [20–26].

Phosphorus removal in the CWS is mainly due to adsorption, complexation, and precipitation processes. The results of longitudinal experiments show that fillings made of natural materials (e.g., gravel, sand, clay) do not ensure effective phosphorus removal in the long run because of their low absorption capacity, which depends on material properties. For instance, research carried out in Denmark [27] showed that the P-binding capacities of sand might vary depending on Ca content.

In the case of industrial waste containing Fe/Al hydrous oxide and $CaCO_3$ (alum water-purification sludge, steel slug, blast furnace slag), the absorption capacity is much higher, which translates into higher P removal effectiveness, compared to sand and gravel [6,19,28]. Furthermore, other media such as vermiculite remove significantly higher amounts of TP (twice the removal efficiency) compared to gravel, which is caused by the adsorption of P on vermiculite surfaces. Vymazal [8,29] states that phosphorus removal in all types of constructed wetland was low unless special media with a high sorption capacity were used. Equally effective, in relation to phosphorus, was manganese oxides [30]. According to the results of a study by Drizo et al. [31] into the physical properties and P adsorption maximum of bauxite, limestone, zeolite, fly ash, and LECA, the highest P adsorption maximum was determined for fly ash (0.86 g/kg), compared to 0.46 g/kg noted for LECA.

A significant increase in the number of installations for thermal treatment sewage sludge makes fly ash management an urgent problem to be solved. The use of fly ash for the production of lightweight aggregates (LWAs), and the use of this material as a filling in CWSs is in line with sustainable development policy and elicits environmental benefits by reducing the consumption of non-renewable natural resources and enabling the reuse of potentially hazardous waste.

Białowiec et al. [32] presented in their publication a product called lightweight aggregates made of fly ash from sewage-sludge thermal treatment (FASSTT LWA). It was prepared following the methodology proposed by Suzuki et al. [33].

The fly ash used in the study was derived from an installation for thermal conversion of sewage sludge of the WWTP facility in Debogorze, Poland. It was highly resistant to pollutant elution, as indicated by 24 h static extraction experiments. Less than 0.1 % of the initial content of C, N, P, Ca, Mg, Zn, Cu, Pb, Ni, Cr, Cd, and Hg were washed out to the solution. The fly ash had high contents of $Ca^{+2}$, $Mg^{+2}$, and $P_2O_5$, and it was potent enough to induce alkaline pH values (3.93 and 2.38% d.m., pH 7.9). The experiments carried out under dynamic conditions demonstrated that the efficiency of phosphate ion removal increased with the decrease in hydraulic loading. The highest value of the total load of removed phosphates (124 mgP/kg FASSTT LWA) was four times higher than the value recorded during experiments performed under static conditions.

The FASSTT LWA filling was also used in vertical-flow-constructed wetlands (VFCWs) treating synthetic wastewater with a mean concentration of 45 mg $N_{NH4}$/L (NH4Cl). A six-month experiment carried out using lysimeters filled with gravel and FASSTT LWA in various proportions and a hydraulic loading rate of 4.67 mm/d demonstrated over 99% ammonia removal efficiency, with nitrogen removal efficiency reaching 59.5% [34].

In turn, Rodziewicz et al. [35] investigated the usability of FASSTT LWA for phosphorus removal from synthetic wastewater containing 7.36 mgP/L. Their study was conducted following the methodology proposed by Białowiec et al. [34], using lysimeters filled with gravel and FASSTT LWA granulate (upper layer), along with hydraulic loading rates of 3, 5, and 7 mm/d. They showed that the phosphorus removal effectiveness depended on the gravel-to-granulate ratio and that phosphorus removal was mainly due to its adsorption on the filling's surface. It is worth mentioning that throughout the study, wastewater was fed to VFCWs once a week.

The properties of FASSTT LWA, including high contents of Ca, Mg, and $P_2O_5$ and the potential to induce alkaline pH values, suggest that filling of this type is potent enough to remove phosphorus. The use of fly ash to produce lightweight aggregates (LWAs) and its further application as a medium in VFCWs filled with FASSTT LW are also examples of actions consistent with sustainable development policy because they elicit environmental benefits by enabling reduced consumption of non-renewable natural raw materials (e.g., sand, gravel, limestone) and reuse of potentially hazardous waste.

In our opinion, Ca and Mg may be involved in phosphorus binding in the form of calcium and magnesium phosphates. What is more is that the presence of $P_2O_5$ and Zn and alkaline pH values may support $Zn_3(PO_4)_2$ precipitation. All the above phenomena may determine the feasibility of using VFCWs with FASSTT LWA for phosphorus removal from wastewater, which may be especially important in rural areas located within agglomerations. Pursuant to the Polish legal regulations from 2019 [36], HSTPs must ensure the removal of phosphorus to the level resulting from agglomeration size. This may refer to the need to remove phosphorus to even less than 1 mg P/L for agglomerations with PE > 100,000. In the case of agglomerations with PE ranging from 10,000 to 99,999 and those with PE from 2000 to 9999, discharging treated wastewater to lakes, their expected effluent phosphorus concentration is 2 mg P/L.

Investigations by Jucherski et al. [37] and Jóźwiakowski et al. [38] demonstrated that HSTP consisting of a septic tank and a trickling filter ensured phosphorus concentration in the effluent at 7.12 ± 3.05 mg P/L. One of the simpler technical solutions to meet the requirements of the regulations mentioned above [36] would be to expand a HSTP with a biofilter/VFCW filled with a material that ensures phosphorus removal, such as FASSTT LWA. Because the wastewater in VFCWs is applied in large doses onto the CW surface, flooding the entire surface, the removal of phosphorus is limited, mainly due to the short and inadequate contact time between the CW media and the wastewater as it flows down by gravity [22]. Therefore, it is necessary to provide an appropriate system for dosing biologically treated sewage, ensuring the extension of the contact time of wastewater with the CW filling. VFCWs have to be preceded by, e.g., a tipping vessel (pump system), enabling even distribution of sewage over the entire surface of the filling and ensuring the use of its entire volume. The tipping vessel could have a volume corresponding to a daily volume of produced sewage, which would enable at least the daily inflow of sewage to the CW.

The aim of this study was to determine whether and how modifying the frequency of wastewater feeding from weekly to daily (at the same weekly dose) would affect the quality of treated wastewater and the efficiency of phosphorus removal from sewage in VFCWs with FASSTT LWA depending on the FASSTT LWA content in the CW filling and hydraulic loading rate. Another goal was to determine whether CWs from FASSTT LWA could be used to expand household sewage treatment plants located in the agglomeration in order to reduce the concentration of phosphorus to a level below 2.0 mg/L and 1.0 mg/L.

The scope of this study included determining the impact of the granulate-to-gravel ratio in the CW and the hydraulic loading rate on the concentration of phosphorus in the effluent and on the efficiency of phosphorus removal from wastewater, as well as comparing the efficiency of phosphorus removal depending on the hydraulic loading rate in the CW filled with FASSTT LWA with that achieved in the CW filled with gravel only.

## 2. Materials and Methods

### 2.1. Lysimeters

This laboratory-scale study was carried out in a greenhouse on constructed wetland with vertical flow, using 15 lysimeters (volume 0.2 m$^3$, height 1 m, diameter 0.5 m) filled with gravel and FASSTT LWA layers in different proportions. Gravel used in the experiment was characterized by a particle diameter $d_{60}$ of 2.1 mm. FASSTT LWA was made from fly ash originating from an SS Incineration Installation, which is part of the Municipal Wastewater Treatment Plant in Gdynia, Poland. Granules of FASSTT LWA had the following parameters: particle diameter of $d_{60}$ 8.2 mm and $d_{60}/d_{10}$ 2.27; squeezing strength of 12.6 N/mm$^2$; bulk density of 0.88 g/cm$^3$; and hydraulic conductivity of $K_h$ 0.52 cm/s [34].

The lysimeters were prepared as double-layer systems. An upper layer was made of FASSTT LWA above the gravel layer with different thicknesses of LWA (CW 0 cm: only gravel; CW 12 cm, CW 25 cm; CW 50 cm, and CW 100 cm: only FASSTT LWA). Each filling variant was repeated three times (Figure 1).

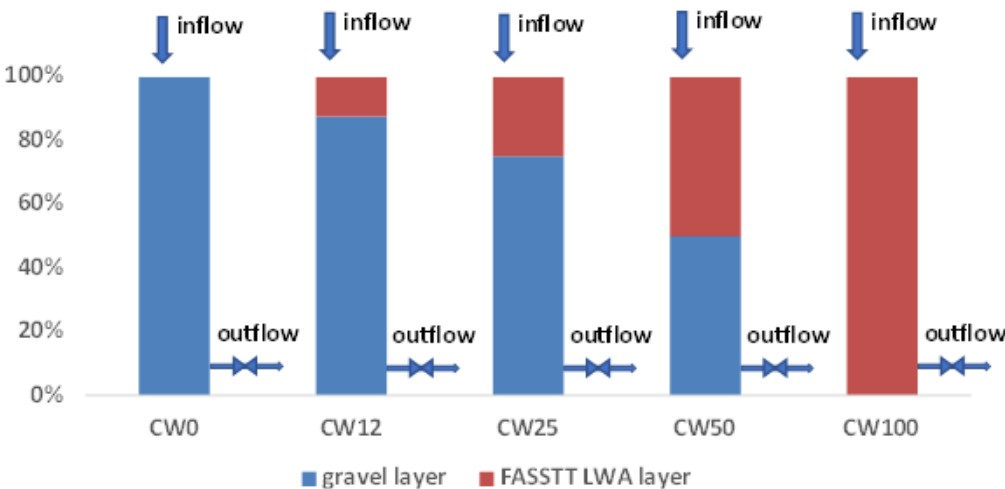

**Figure 1.** Experimental scheme of equipment.

The valves for wastewater feeding and for the collection of effluent samples for analyses were mounted 3 cm above the lysimeters (Figure 2).

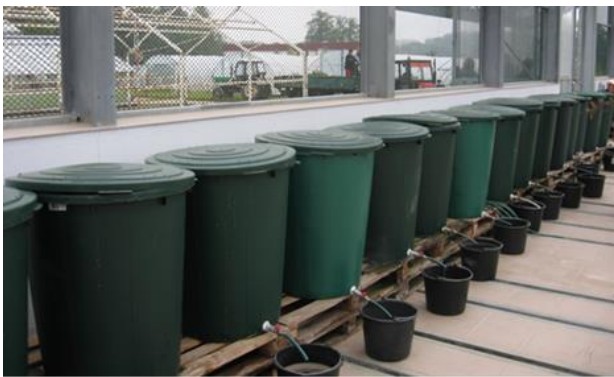

**Figure 2.** Test stand with lysimeters.

*2.2. Experimental Procedure*

Experiments were performed in a greenhouse at an average air temperature of 23.2 °C (±3.9) for 13 weeks. Wastewater was fed to the VFCW for 30 days prior to the actual investigation and analytical control of the processes. Lysimeters were operated with a vertical flow of synthetic wastewater. It was prepared using the modified methodology adopted by Kasprzyk and Gajewska [39]. A solution of dipotassium hydrogen phosphate $KH_2PO_4$ and tap water (instead demineralized water) was applied, as well as hydraulic loading rates of 3.0, 5.0, and 7.0 mm/d. The assumed loadings were in the range of low hydraulic loading values used in the VFCW [6,40]. The mean phosphorus concentration in raw wastewater was 7.43 mg P/L. The influent had pH = 7.71 and a temperature of 20.28 °C (±0.91) (mean values from the entire experimental period).

After the effluent was discharged through the intake valve, another dose of raw wastewater was fed to the system. It was fed to the CW every day and spread evenly over the filling's surface.

Once a week, the influent and effluent samples were determined for the physicochemical properties, namely phosphorus concentration, temperature, and pH value, as follows:

- Total phosphorus (measurement accuracy to 0.01 mgP/L) using a UV-VIS 5000 DR spectrophotometer (HACH Lange, Düsseldorf, Germany) with the HACH Lange LCK 348–350 method;
- pH value (measurement accuracy to 0.01 pH) and temperature (exact to 1 °C), measured using a CP-105 pH meter (Elmetron, Zabrze, Poland).

## 3. Results and Discussion

The concentration of phosphorus in the influent ranged from 6.74 to 8.24, with the mean value reaching 7.43 mg P/L (Figure 3).

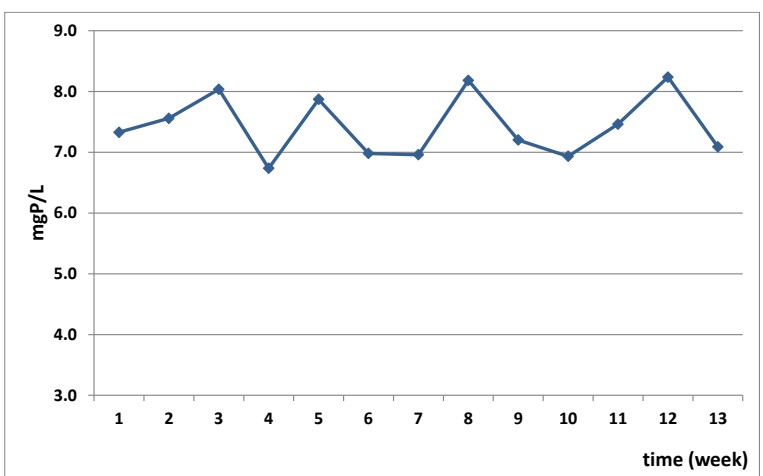

**Figure 3.** Phosphorus concentration in the influent.

The mean temperature of the effluent over the study period reached 22.9 °C (±0.5), whereas its final pH values ranged from 7.47 (100% of gravel) to 9.39 (100% of FASSTT LWA). According to Rahman et al. [6], both parameters were favorable for purification processes.

Figures 3–5 depict the effect of the hydraulic loading rate on phosphorus concentration in the effluent depending on FASSTT LWA content in the CW filling. At a hydraulic loading rate of 3 mm/d, the phosphorus concentration in the effluent decreased along with an increasing granulate content in the CW filling (Figure 3). In the case of CW 12, the phosphorus concentration in the effluent was insignificantly lower compared to that noted for the CW filled in 100% with gravel, i.e., 2.19 ± 0.35 and 2.02 ± 0.14 mg P/L.

Increasing granulate content in the filling of lysimeters to 25% caused a decrease in effluent concentration to less than 2.0 mg P/L ($1.83 \pm 0.24$ mg P/L). In the case of the subsequent lysimeters CW 50 and CW 100, the phosphorus concentration in the effluent dropped below 1.5 mg P/L and below 1.0 P/L, reaching $1.13 \pm 0.20$ mg P/L and $0.97 \pm 0.18$ mg P/L, respectively.

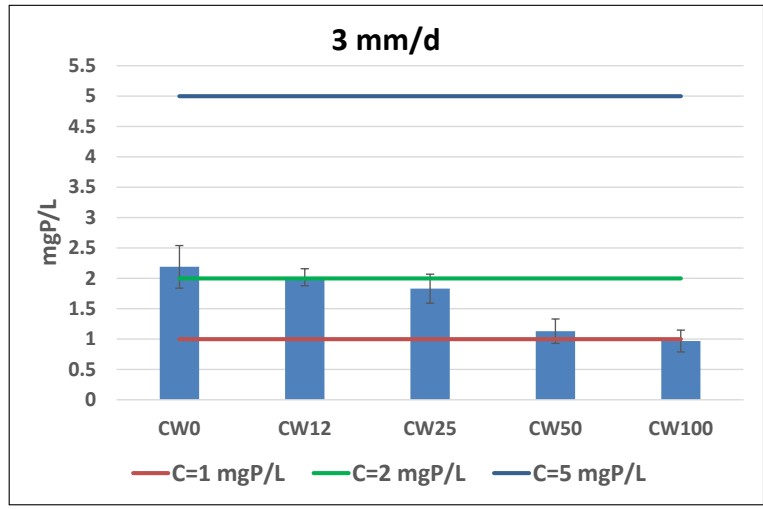

**Figure 4.** Effect of FASSTT LWA content in CW on the phosphorus concentration in the effluent at a hydraulic loading rate of 3 mm/d.

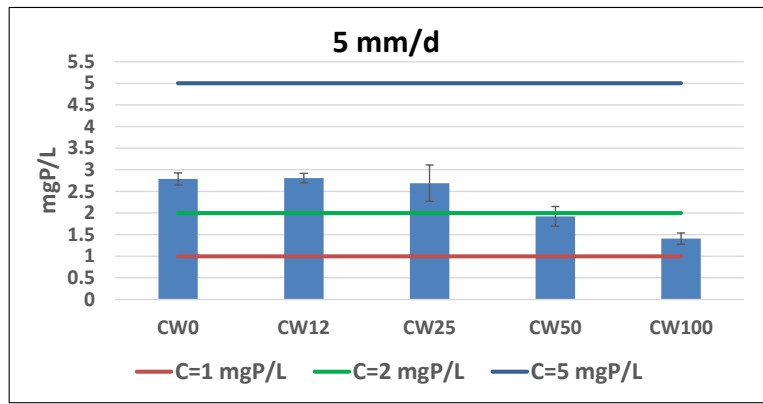

**Figure 5.** Effect of FASSTT LWA content in CW on the phosphorus concentration in the effluent at a hydraulic loading rate of 5 mm/d.

At a hydraulic loading rate of 5 mm/d (Figure 4), the phosphorus concentration in the effluent was higher than 2.5 mg P/L for the first three lysimeters, namely CW 0, CW 12, and CW 25. The phosphorus concentration in the effluent in the lysimeter filled with gravel was comparable with that noted in CW 12, i.e., $2.79 \pm 0.14$ and $2.81 \pm 0.11$ mg P/L, respectively, whereas it was negligibly lower in the effluent from CW 25 and reached $2.69 \pm 0.42$ mg P/L. The phosphorus concentration in the effluent from CW 50 was below 2.0 mg P/L ($1.92$ mg P/L $\pm 0.20$) and that in the effluent from CW 100 was below 1.5 mg P/L ($1.41 \pm 0.13$ mg P/L).

The phosphorus concentrations in the effluents determined at a hydraulic loading rate of 5 mm/d were higher than at a hydraulic loading rate of 3 mm/d (Figure 5).

Similar tendencies were observed for phosphorus concentrations in wastewater treated in lysimeters at a hydraulic loading rate of 7 mm/d (Figure 6). They were higher than at 5 mm/d and significantly higher than at 3 mm/d. Only in the case of CW 100 was the phosphorus concentration lower than 2.0 mg P/L ($1.62 \pm 0.38$ mg P/L).

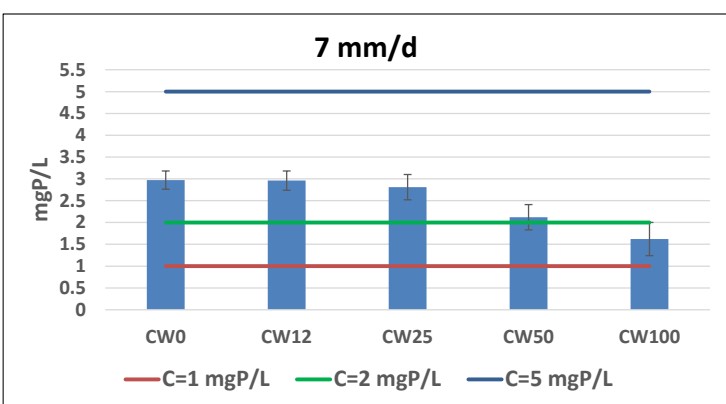

**Figure 6.** Effect of FASSTT LWA content in CW on the phosphorus concentration in the effluent at a hydraulic loading rate of 7 mm/d.

This study demonstrates that in the case of the CW with the filling made of 100% FASSTT LWA, the phosphorus concentration in the effluent was below 1 mg P/L at the lowest hydraulic loading tested, below 1.5 mg P/L at $q_h$ = 5 mm/d, and below 2.0 mg P/L at $q_h$ = 7 mm/d. In the case of the fillings made of gravel, or those that were a mixture of granulate and gravel, the phosphorus concentration in the effluent was higher compared to that determined for CW 100. This proves that there is no technological nor environmental justification for the use of gravel + FASST LW mixtures in constructed wetlands. Given that the FASST LWA filling is made of fly ash, which is a type of waste, the adsorption properties of the produced granulate make it a fine alternative to commercial fillings, such as Filtralite, HelioFIR, Norlite, Phoslock, and LECA [19], and especially to those made of non-renewable natural materials.

The phosphorus removal efficiencies determined throughout the experiment in the CW fed with wastewater on a daily basis (Figure 7) differed from those recorded in the lysimeters fed with wastewater once a week [34]. A similar methodology was applied in the present study to the above-mentioned work (wastewater with a similar phosphorus concentration, the same hydraulic loading rates, the same filling, and similar temperature conditions), yielding comparable results in both. In the cited study [34], no tendency was observed that would indicate an increase phosphorus removal efficiency along with an increase in granulate content in the CW filling. There was also no removal efficiency decrease along with the hydraulic loading rate increase. For all fillings tested, the lowest phosphorus removal efficiency was demonstrated at a hydraulic loading rate of 5 mm/d (53.3% for CW 12). In contrast, the highest efficiency was noted in the lysimeter with CW 50 filling; it reached 87.7% at a hydraulic loading rate of 3 mm/d [34]. Opposite observations were made in the present study. Although the differences between phosphorus removal efficiencies in the lysimeter with gravel (CW 0) and in the lysimeter with a 12 cm granulate layer (CW 12) were small, especially at hydraulic loading rates of 5 and 7 mm/d (Figure 7), in the case of the subsequent fillings, the removal efficiencies increased noticeably, along with an increase in granulate content in the lysimeter filling.

Therefore, at $q_h$ =3 mm/d, phosphorus removal efficiency reached 75.4% for CW 25 and grew to 86.9% for CW 100. In the case of hydraulic loading rates of 5 and 7 mm/d, the respective values were at 63.8% and 81.0%, as well as 62.2% and 78.2%, indicating a noticeable increase in phosphorus removal efficiency due to both an increasing granulate content in the filling and a decreasing hydraulic loading rate. The highest efficiency, reaching 86.9%, was determined for CW 100 at a hydraulic loading rate of 3 mm/d. This is lower than the values obtained by Kasprzyk and Gajewska [39]. In their studies, the efficiency of phosphorus removal ranged from 96–99%, depending on the type of filling (calcium oxide and lanthanum-modified bentonite) used and the contact time of synthetic sewage with the filling. It should be noted that the wastewater used in these studies had a concentration of phosphorus twice as high (15.0 and 7.43 mg P/L, respectively).

According to the authors, a concentration of 15.0 mg P/L is close to the concentration of wetland effluent, whereas a concentration of 7.43 mg P/L is close to secondary settling tank, following the trickling filter and effluent. The method of contact of sewage with the filling was also clearly different. In research conducted by Kasprzyk and Gajewska [39] the mechanical mixing of the filling with wastewater in the beakers was applied. In our own research, in lysimeters, the contact of sewage with the filling was similar to that in the VFCW. Additionally, in research conducted by Gubernat et al. 2023 [41]. with the use of raw marl, raw travertine, heated marl, heated travertine, and Polonite® treating $KH_2PO_4$ solutions with a concentration of 19.87 mg P/L, the efficiency of phosphorus removal for Polonite and raw marl media was lower for the majority of fill doses than in the studies discussed in this article. The obtained efficiency of wastewater treatment for CW 100 filling for all tested hydraulic loads was higher than 75% and clearly exceeded the values observed in tests conducted by Chai et al. [42] in barrels with a biological ceramist and with no plants. The subject of the research was rainwater from harvesting system tanks with a phosphorus concentration of COD below 40 mg $O_2$/L (similar to tap water COD value) and phosphorus concentration of up to 0.15 P/L.

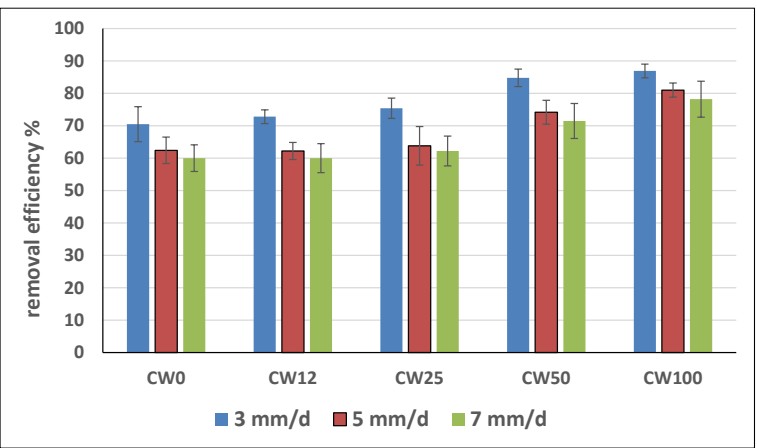

**Figure 7.** Phosphorus removal efficiency.

It is worth noting that the phosphorus removal efficiency in CW 100 at the hydraulic loading rates tested (i.e., 7, 5, and 3 mm/d) ranged from 78.2 to 86.9%, which ensured phosphorus concentration in the effluent below 2.0 mg P/L (Figures 4–6). This means that the effluent from the CW with FASSTT LWA met the requirements of the HELCOM Recommendation 28E/6, which advises that at least 70% phosphorus removal from wastewater discharged from the HSTPs [43]. Simultaneously, the efficiency of phosphorus removal in the lysimeter filled with gravel ranged from 60.0 to 70.5%. These differences are due to the fact that on the filling made of sand and gravel, phosphorus is mainly bound via adsorption and precipitation with calcium (Ca), aluminum (Al), and iron (Fe). In turn, the contents of these ions in gravel are significantly lower than in FASSTT LWA [32]. At pH > 6.0, phosphorus removal proceeds not only via physical adsorption on iron and aluminum ions and precipitation in the form of sparingly soluble calcium phosphates. The influent fed to lysimeters had pH = 7.71, whereas the pH of the effluent ranged from 7.47 (100% of gravel) to 9.39 (100% of FASSTT LWA), which facilitated more intensive phosphorus removal in the process of its precipitation from the CW with FASSTT LWA. Many authors [44–46] have emphasized that alkaline conditions promote calcium phosphate precipitation. In the present study, medium alkalinity in CW was observed to increase, along with an increase in the percentage content of the granulate.

The positive impact of the addition of alkaline fly ash on phosphorus removal efficiency has already been reported by Cheung and Venkitachalm [47] and Kim et al. [48]. Reaching over 85% phosphorus removal efficiency on the natural filling (sand) is feasible

on the condition that it has a high calcium content and is used in horizontal-constructed wetlands [49] that are overgrown with vegetation [50].

The phosphorus removal efficiencies noted in CW 100 were lower than the values reported by Jucherski et al. [5] in the range of 93.2 to 94.9% for wastewater with a similar mean phosphorus concentration. This difference is due to the use of specialist fillings (Rockfos, Rockfos + Leca mixture) and chemical composition of the materials used [51,52], as well as different technological and hydraulic regimes. In the cited study, raw wastewater was fed to columns with filtration materials in a continuous mode, whereas the filling was overflown with the influents, and HRT was 12 h. In the present experiment, wastewater was fed periodically, the real contact time of wastewater with the filling was limited owing to the short wastewater flow through the filling during sprinkling, and a small volume of this part of the filling was in contact with wastewater for 24. The contact time between the filling and the loading of the filling surface with phosphorus (the load referred to as the total specific surface area) is one of the major factors determining adsorption effectiveness [53]. The above finding has also been confirmed in a study [54] showing that the formation of amorphous calcium phosphate during calcium phosphate precipitation depends on reaction time and temperature.

The present experiment was performed in a greenhouse under favorable temperature conditions, i.e., the mean temperature of the effluent reached 20.28 °C ($\pm$0.91). For this reason, we did not note a dramatic diminishment in effluent quality compared to that reported by Jucherski et al. [5]. The precipitation of calcium phosphates is an endothermal reaction, the rate of which increases along with temperature increase [55], meaning that precipitation proceeds faster and more effectively at higher temperatures [56,57].

The present study results demonstrate that CWs filled with FASSTT LWA can be used during the third stage of the treatment process of wastewater after its bio-treatment in HSTPs operating based on, e.g., a septic tank and a trickling filter, on the condition that the wastewater is fed daily by means of a tipping vessel or a pumping station. As shown in the present study, the treatment of wastewater of this type, i.e., having a low content of suspended matter, high contents of nitrates, and significant loads of phosphorus, on CWs with FASSTT LWA may decrease phosphorus concentration in the effluent below 2.0 mg P/L. This is confirmed by the results of Verma et al. [58], according to which suitable pre-treatment may increase the applicability of CWs for phosphate removal from domestic wastewater. Interestingly, this level of P concentration can be achieved in the entire tested range of hydraulic loading rates, i.e., from 3.0 to 7.0 mm/d. This increases the reliability of the system based on the CW with FASSTT LWA, and the expected treatment efficiency is reached even when the assumed values of hydraulic loading are exceeded.

Considering the results of the previous experiments conducted by the authors of this work [34,35], as well as findings from the present study and those reported by Jucherski et al. [5], we are aware of the feasibility of boosting phosphorus removal efficiency in VFCWs with FASSTT LWA. This efficiency increase can be achieved by complete over-flowing of the filling with the influent and increased frequency of influent feeding at the maintained hydraulic loading rate used so far. The first mentioned treatment extends wastewater contact time with the filling and increases the surface of pollutant contact with FASSTT LWA. It also contributes to more evenly distributed, and thus more extended, CW loading in time with pollutants. However, further studies are needed to confirm these speculations.

## 4. Conclusions

The present study results demonstrate that modifying wastewater feeding frequency from a weekly to daily manner contributes to increased phosphorus removal efficiency in vertical-constructed wetlands with aggregates made of fly ash from sewage-sludge thermal treatment. This was mainly due to high contents of Ca, Al, and Fe in fly ash used to produce FASSTT LWA and alkaline conditions occurring in CW. Both the increasing percentage content of the granulate (FASSTT LWA) in the CW filling and a decreasing

hydraulic loading rate were found to reduce phosphorus concentration in the effluent. Effluent with the lowest concentration of phosphorus was recorded in the case of the CW filled with 100% FASSTT LWA for all applied hydraulic loading rates. Constructed wetlands filled with 100% FASSTT LWA ensured phosphorus concentration reduction below 2.0 mg P/L at all hydraulic loading rates tested (3, 5, and 7 mm/d). Wastewater with the lowest concentration of phosphorus, below 1 mg P/L, was discharged from the VFCW with a hydraulic loading rate of 3 mm/d. Phosphorus removal efficiencies achieved in the lysimeters with the granulate (78.2 to 86.9%) were noticeably higher than those determined in the lysimeters filled in 100% with gravel (CW 0), i.e., 60.0–70.5%. This confirms the thesis that a change in the frequency of wastewater supply with a simultaneous reduction in a single dose increases the time of contact of sewage with the filling and, consequently, increases the efficiency of the adsorption, complexation, and precipitation processes. It can be expected that dividing the daily dose of wastewater into smaller doses and multiplying the supply of wastewater to the VFCW results in even higher efficiency when removing phosphorus compounds.

VFCWs filled with FASSTT LW can aid HSTPs based on a septic tank and a device for simplified wastewater treatment (such as a trickling filter or an activated sludge tank). Such a technological system will meet the requirements set in the HELCOM Recommendation for wastewater treatment plants designed for PE < 300 and will be applicable in agglomerations with PE ranging from 10,000 to 99,999, as well as in those with PE from 2000 to 9999, discharging the effluent to lakes and their tributaries.

**Author Contributions:** Conceptualization, W.J.; methodology, J.R., A.B. and A.M.; validation, J.M.R.T. and K.J.; formal analysis, W.J., J.R. and A.M.; investigation, J.R., W.J. and A.M.; resources, W.J. and J.R.; data curation, J.R.; writing—original draft preparation, J.R., W.J., A.M., A.B., A.T., J.M.R.T. and K.J.; writing—review and editing, J.R. and W.J.; visualization, J.R. and J.M.R.T.; supervision, W.J. and A.T.; project administration, J.R. and A.M.; funding acquisition, J.R., A.M. and W.J. All authors have read and agreed to the published version of the manuscript.

**Funding:** Project financially supported by the Minister of Education and Science under the program entitled "Regional Initiative of Excellence" for the years 2019–2023, Project No. 010/RID/2018/19, amount of funding 12.000.000 PLN. The study was financially co-supported in the framework of a Project no. 29.610.023-300 of the University of Warmia and Mazury in Olsztyn, Poland.

**Data Availability Statement:** The data presented in this study are available on request from the corresponding author.

**Acknowledgments:** The article was created under Project no. POWR.03.05.00-00-Z310/17: "Development Program of the University of Warmia and Mazury in Olsztyn".

**Conflicts of Interest:** The authors declare no conflict of interest. The funders had no role in the design of the study; in the collection, analyses, or interpretation of data; in the writing of the manuscript, or in the decision to publish the results.

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
