# Peer review of "Phosphorus Removal in VFCWs with Lightweight Aggregates Made of Fly Ash from Sewage-Sludge Thermal Treatment (FASSTT LWA)"

_water, doi:10.3390/w15101955_

Round 1
Reviewer 1 Report (New Reviewer)
I found this article very interesting.
In my opinion the manuscript is well organized. The problem statements agree with the title and have significance. The methods used to gather the data for this article were clearly explained. The quality of citations is good (because in this moment publish a lot of new articles), autors have referenced the interesting works in this field of research. The topic is interested and the result are concreted and useful for the scientific community.
My only remark concerns too large spacing in the text, e.g. in lines 444, 367.
Conclusions presented in a concise form.
In my opinion, this is a very valuable publication
and is a valuable source of information and forms the basis for further analyzes and implementation of new solutions.
Thank you for considering my opinion. I encourage the authors to continue working on improving the manuscript.
Author Response
Dear Reviewer 1,
Thank You very much for Your comments.
These are our answers:
I found this article very interesting.
In my opinion the manuscript is well organized. The problem statements agree with the title and have significance. The methods used to gather the data for this article were clearly explained. The quality of citations is good (because in this moment publish a lot of new articles), authors have referenced the interesting works in this field of research. The topic is interested and the result are concreted and useful for the scientific community.
My only remark concerns too large spacing in the text, e.g. in lines 444, 367.
We corrected it.
Conclusions presented in a concise form.
In my opinion, this is a very valuable publication
and is a valuable source of information and forms the basis for further analyzes and implementation of new solutions.
Thank you for considering my opinion. I encourage the authors to continue working on improving the manuscript.
Dear Reviewer 1 thank you again for Your comments
Reviewer 2 Report (New Reviewer)
Dear authors,
Thank you for submitting your scientific article for review. I have read through your paper carefully and have the following remarks:
On L187, I would appreciate more information on how "Wastewater" was obtained, how it was stored, and what its chemical composition was. This information would be helpful for better understanding the context of your study.
There are some minor typographical errors on L191 that need to be corrected for clarity.
While you indicated the method for total P measurement on L202, I noticed on L226-242 that you talked about PPO4, but not total P. Could you please clarify how the total P was determined in your study?
The methodology section is incomplete, as there is no technological/experimental scheme of equipment. Including this information would make it easier for readers to understand the experimental setup.
I also noticed that the self-citation ratio in your paper is quite high, with at least eight self-cited references. While it is important to acknowledge your previous work, I would suggest that you consider including more external references to support your arguments.
Overall, I appreciate the effort put into this research and believe that it has the potential to contribute to the scientific community. With these revisions, I believe your paper could be improved and would be more informative for readers.
There are typographical mistakes.
Author Response
Dear Reviewer 2,
Thank You very much for Your comments.
These are our answers:
On L187, I would appreciate more information on how "Wastewater" was obtained, how it was stored, and what its chemical composition was. This information would be helpful for better understanding the context of your study.
Wastewater was prepared every day. It was prepared using a modified methodology adopted by Kasprzyk and Gajewska (Phosphorus removal by application of natural and semi-natural materials for possible recovery according to assumptions of circular economy and closed circuit of P. Sci. Total Environ. 2019, 650, 249-256. https://doi.org/10.1016/j.scitotenv.2018.09.034).
A solution of potassium dihydrogen phosphate KH2PO4 and tap water were applied. After the effluent was discharged through the intake valve, another dose of raw wastewater was fed to the system. It was fed to the CW every day and spread evenly over the filling's surface. The mean phosphorus concentration in raw wastewater was 7.43 mg P/L. The influent had pH=7.71, COD was comparable with COD of tap water.
There are some minor typographical errors on L191 that need to be corrected for clarity.
We corrected it.
While you indicated the method for total P measurement on L202, I noticed on L226-242 that you talked about PPO4, but not total P. Could you please clarify how the total P was determined in your study?
The value of total phosphorus is given throughout the experiment. Phosphorus in the form of orthophosphates was introduced into the raw wastewater by using potassium dihydrogen phosphate (KH2PO4). The method of total phosphorus determination during the experiment was performed using a UV-VIS 5000 DR spectrophotometer with the HACH Lange LCK 348–350 method, taking into account the mineralization of the sample. Mineralization of the sample caused all forms of phosphorus to convert to orthophosphates (PO43-). The value has been given in terms of pure phosphorus, i.e. P. We changed that in whole article replacing mg PPO4/L with mg P/L.
The methodology section is incomplete, as there is no technological/experimental scheme of equipment. Including this information would make it easier for readers to understand the experimental setup.
We added new figure with technological scheme of an experiment and information about lysimeter’s variants.
I also noticed that the self-citation ratio in your paper is quite high, with at least eight self-cited references. While it is important to acknowledge your previous work, I would suggest that you consider including more external references to support your arguments.
We have removed two articles of our co-authorship. We can't do more because we need them to discuss the results and show the previous research that inspired us to write this article.
Now there are 6 of our articles out of 58 listed in the References chapter.
Overall, I appreciate the effort put into this research and believe that it has the potential to contribute to the scientific community. With these revisions, I believe your paper could be improved and would be more informative for readers.
Comments on the Quality of English Language: There are typographical mistakes.
We corrected it.
Dear Reviewer 2 thank you again for Your comments
Reviewer 3 Report (New Reviewer)
The authors of the manuscript continue their research, the results of the previous ones are presented in the references. The current work is devoted to the evaluation of the influence of the filling composition and loading rates on the efficiency of phosphorus removal. An important feature of this study is the use of waste (FASSTT LWA) as a filling material. There are a few comments that may help improve the quality of the presented manuscript.
Proofreading check is required.
Perhaps figures 2-5 need to be improved
Lines 190-191. Either the name or the formula for phosphate is incorrectly used. What is the rationale for choosing this particular salt and is there any effect of using others on the results presented below? An explanation is required.
Lines 191-192. What is the reason for choosing such low hydraulic loading rates? Only 0.6, 1 and 1.4 liters per day pass through the 200-liter lysimeter. Wastewater hit the lysimeter at one point or evenly over the area? Additional clarification is required.
Fig. 2. Lines 208-209. What caused the change in the concentration of phosphorus in the influent in such a range, explanations are needed.
Author Response
Reviewer 3
Dear Reviewer 3 thank You very much for Your comments.
These are our answers:
The authors of the manuscript continue their research, the results of the previous ones are presented in the references. The current work is devoted to the evaluation of the influence of the filling composition and loading rates on the efficiency of phosphorus removal. An important feature of this study is the use of waste (FASSTT LWA) as a filling material. There are a few comments that may help improve the quality of the presented manuscript.
Proofreading check is required.
We checked the article.
Perhaps figures 2-5 need to be improved
We improved them.
Lines 190-191. Either the name or the formula for phosphate is incorrectly used. What is the rationale for choosing this particular salt and is there any effect of using others on the results presented below? An explanation is required.
The correct name of KH2PO4 - potassium dihydrogen phosphate. This compound, in addition to providing phosphorus, also provides potassium necessary for the development of biomass. In addition, potassium dihydrogen phosphate has buffering properties, stabilizing the pH, without causing acidification of the wastewater prepared in this way.
We made changes in whole article replacing mg PPO4/L with mg P/L.
Lines 191-192. What is the reason for choosing such low hydraulic loading rates? Only 0.6, 1 and 1.4 liters per day pass through the 200-liter lysimeter. Wastewater hit the lysimeter at one point or evenly over the area? Additional clarification is required.
The applied and tested hydraulic loadings (3-7 mm/d) turned out to be accurate, because they guaranteed the effectiveness of wastewater treatment to the extent required by applicable regulations. CW is a solution used in rural areas, where in most cases there are no restrictions. Under these conditions, the size of the hydraulic load and the resulting CW area are secondary issues compared to the technological effects achieved and minimal energy expenditure.
In the chapter “Materials and Methods, 2.2. Experimental procedure”, we quoted literature items showing that similar values of hydraulic loads (0.006 m/d) for VF CW had already been used. (Alexandros Stefanakis, Christos S. Akratos, Vassilios A. Tsihrintzis; Vertical Flow Constructed Wetlands Eco-engineering Systems for Wastewater and Sludge Treatment, 2014).
Wastewater was fed to the CW every day and spread evenly over the filling's surface.
Fig. 2. Lines 208-209. What caused the change in the concentration of phosphorus in the influent in such a range, explanations are needed.
We wanted to create conditions similar to those that take place in real sewage treatment plants, where the concentration of treated wastewater fluctuates. Hence the variability of the inflow concentration (±10%), but within a reasonable range 6.74 to 8.24, mg P/L. And this is supposed to be the purpose of VF biofilters with FASSTT LWA filling, to remove phosphorus from wastewater previously subjected to biological treatment in a settling tank and a trickling filter.
Dear Reviewer 3 thank you again for Your comments
Round 2
Reviewer 3 Report (New Reviewer)
Accept
This manuscript is a resubmission of an earlier submission. The following is a list of the peer review reports and author responses from that submission.
Round 1
Reviewer 1 Report
The work deals with a current topic with realistic results.
To be specified in a legend for figures 3, 4 and 5 which represent the red, green, blue lines (represented on the graphs).
Author Response
Dear Reviewer 1,
Thank You very much for Your comments.
These are our answers:
To be specified in a legend for figures 3, 4 and 5 which represent the red, green, blue lines (represented on the graphs).
They have been added in the legend, meaning the red, green, and blue lines in Figures 3, 4, and 5.
Dear Reviewer 1
Thank You very much for Your comments.
Reviewer 2 Report
The paper reports the results of a study where a mixture of gravel and aggregates coming from fly ashes is used as a filter medium for the phosphorous removal from wastewater which already received a biological treatment. The paper is in line with the aims and scope of the journal but I found it of limited interest for the journal’s readers. Here a list of the main weaknesses of the paper:
The tested apparatus is not a constructed wetland because plants/vegetation are not present.
The introduction is definitely too long and does not focus on the crucial point of the paper, that is, are the tested hydraulic loading rates (around 5 mm/d) realistic for this kind of systems? Being in mind that these systems are filters and not wetlands. Given this HLR, how wide should be the surface of the filter for each equivalent inhabitant?
Most of the references dates back to before 2015 and the results are mainly discussed with reference to a previous study by the same authors (ref. n. 29 in Polish). For what concerns the other studies used to support the discussion, it is not clear if the conditions used to carry out those studies are similar to those of this study.
The authors mentioned (line 136) a possible role of Zn in P removal, Zn was added to the synthetic wastewater but, actually, its effect was not elucidated. How where the concentrations of P, Zn and NH4 chosen? Why COD was not present?
Aggregates were obtained from fly ashes which were characterized in 2009, is that possible?
Has the contact between such aggregates and the eluting wastewater an effect on the quality of the treated wastewater? (I mean: can those aggregate release some metals?)
Considering the activities carried out in this study (one measure of temperature, P and pH once a week, no advanced elaboration of the data), the number of authors seems quite high.
Author Response
Dear Reviewer 2,
Thank You very much for Your comments.
These are our answers:
- The tested apparatus is not a constructed wetland because plants/vegetation are not present.
The subject of the article concerns the removal of phosphorus compounds only. Nitrogen removal issue in vertical flow constructed wetlands containing LWA/gravel layers and reed vegetation were presented in an earlier article (BiaÅ‚owiec, A.; Janczukowicz, W.; Randerson, P.F. Nitrogen removal from wastewater in vertical flow constructed wetlands containing LWA/gravel layers and reed vegetation. Ecol. Eng. 2011, 37, 897–902). Phosphorus removal is mainly due to adsorption, complexation, and precipitation processes in CW media. The results of longitudinal experiments show that it mainly depends on media material properties. In vertical wetlands the removal of P is limited, mainly due to inadequate contact time between the porous media and the wastewater as later flows down by gravity (Brix H., Arias C.A, The use of vertical flow constructed wetlands for on-site treatment of domestic wastewater: New Danish guidelines, Ecological Engineering December 2005...; Stefanakis A., Tsihrintzis V., Effects of loading, resting period, temperature, porous media, vegetation and aeration on performance of pilot-scale vertical flow constructed wetlands, Chemical Engineering Journal, V 181–182, 1 February 2012, Pages 416-430). That’s why we did not plant CW with reed.
In the light of the literature, the presence of plants in VFCW would not have a significant effect on the removal of phosphorus.
- The introduction is definitely too long and does not focus on the crucial point of the paper, that is, are the tested hydraulic loading rates (around 5 mm/d) realistic for this kind of systems? Being in mind that these systems are filters and not wetlands. Given this HLR, how wide should be the surface of the filter for each equivalent inhabitant?
The introduction has been shortened.
In our article (lines: 93-130) we quoted our earlier studies on the granulate itself (position [26]), nitrogen removal efficiency in lysimeters filled with granulate and gravel (positions [28] and [29].
Articles [28] and [29] provide the values of the assumed hydraulic loading rates. The first publication describes research on the removal of ammonium nitrogen in which the hydraulic loading rate was about 5 mm/d (4.67mm/d) which is characterized for low loaded nitrifying bed.
In the studies presented in publication [29], we used hydraulic loading rates of 3, 5 and 7 mm/d, but the sewage was dosed once a week. Here, the concentration of phosphorus in the treated wastewater was 7.36 mg PPO4/L. In the research that are the subject of our publication, the mean phosphorus concentration in raw wastewater was 7.43 mg PPO4/L. We really wanted these values to be similar, which was achieved thanks to the use of synthetic sewage. Investigation [29] showed that the phosphorus removal effectiveness depended on the gravel-to-granulate ratio and that phosphorus removal was mainly due to its adsorption on filling's surface. In the case of the 3 mm/d loading rate, the phosphorus removal efficiency ranged from 58.2 % to 87.7% depending on the filling of the granulate.
Research by Professor Jóźwiakowski et al., one of the co-authors of our article [32, 33], demonstrated that a household sewage treatment plant consisting of a septic tank and a trickling filter ensured phosphorus concentration in the effluent at 7.12 ± 3.05 mg PPO4/L.
All this led us to jointly decide to conduct research to determine the possibility of using VFCW with FASSTTLWA to remove phosphorus from biologically treated wastewater [32, 33], so that the outflow meets the requirements for wastewater discharged from HSTPs located in agglomerations. Research has confirmed that it is possible for CW 100 (100% of FASSTTLWA). The level of P concentration below 2.0 mg P/L can be achieved in the entire tested range of hydraulic loading rates, i.e., from 3.0 to 7.0 mm/d, whereas below 1.0 mg P/L just only for 3 mm/d.
If unit average day flow equals 150 L/M· d, then we need the surface of 30 m2 for 1 pe.
- Most of the references dates back to before 2015 and the results are mainly discussed with reference to a previous study by the same authors (ref. n. 29 in Polish). For what concerns the other studies used to support the discussion, it is not clear if the conditions used to carry out those studies are similar to those of this study.
It is natural that if we carry out comparative studies with the results of the experiment carried out in 2015, maintaining similar conditions and changing the method of dosing wastewater, we refer the obtained results to the results presented in the publication from 2016.
At the same time, when discussing our results, we also refer to publications published after 2015 on the same topic, i.e., phosphorus removal. Rarely is there such comfort as in the case of our research. The thesis formulated on the basis of the literature review could be verified by us by direct comparison with the results of the research from 2016.
- The authors mentioned (line 136) a possible role of Zn in P removal, Zn was added to the synthetic wastewater but, actually, its effect was not elucidated. How where the concentrations of P, Zn and NH4 chosen? Why COD was not present?
The synthetic wastewater (prepared with tap water) contained only phosphorus compounds. The COD was low, characteristic of tap water, as was the concentration of Zn (galvanized steel pipes are used in Poland in water supply system and in plumbing), which must meet the values for drinking water. Zinc could also come from aggregates, but due to the low elution rate, it could not be too much (according to [26]). Concentration of P was chosen in the first place based on research of Professor Jóźwiakowski et al., one of the co-authors of our article [32, 33], and our previous investigation [29], because we wanted to be able to compare the quality of the treated wastewater at the VFCW, which was fed once a week. Therefore, tap water was used to prepare the synthetic wastewater so that the concentration of zinc in the inflow to the VFCW was close to the concentration of Zn in the wastewater flowing out of household wastewater treatment plants, where its removal is severely limited.
- Aggregates were obtained from fly ashes which were characterized in 2009, is that possible?
Fly ashes were examined and described in detail in 2009. In the following years, we used aggregates made from these ashes. The result was further research and publications.
- Has the contact between such aggregates and the eluting wastewater an effect on the quality of the treated wastewater? (I mean: can those aggregate release some metals?)
In paper [26] we characterized lightweight aggregates made from fly ashes coming from sewage sludge thermal treatment. High resistance of pollutant elution from fly ashes was determined during 24 hours of extraction less than 0.1% of the initial content of analyzed elements (C, N, P, Ca, Mg, Zn, Cu, Pb, Ni, Cr, Cd, and Hg) had been washed out into solution. Taking into account the time of contact of wastewater with the filling and the form of this contact, an even lower release of metals to the treated wastewater can be expected.
- Considering the activities carried out in this study (one measure of temperature, P and pH once a week, no advanced elaboration of the data), the number of authors seems quite high.
Explanation regarding the number of authors and their role in initiating and planning research is presented in the second part of point 2 of our answers.
Dear Reviewer 2
Thank You very much for Your comments.
Reviewer 3 Report
TITLE - seems interesting; refers to the content of the manuscript; this way the title is too long and too detailed; it should be shortened.
INTRODUCTION - is too long, it can be shortened.
MATERIALS AND METHODS - has little novelty and is already used in similar investigations.
RESULTS AND DISCUSSION - are satisfactory; the description of figures is clear and understandable.
CONCLUSIONS - are general. The innovativeness of the conducted research was not emphasized.
REFERENCES - are adequate, correct and in line with the previous chapters. The use of the bibliography is correct
Recommendation
I recommend this manuscript for publication in the journal Water with minor corrections.
Author Response
Dear Reviewer 3,
Thank You very much for Your comments.
These are our answers:
TITLE - seems interesting; refers to the content of the manuscript; this way the title is too long and too detailed; it should be shortened.
The Title has been shortened.
INTRODUCTION - is too long, it can be shortened.
The INTRODUCTION has been shortened.
MATERIALS AND METHODS - has little novelty and is already used in similar investigations.
The idea behind the research is to check whether the previously tested, in many respects, filling can be used to remove phosphorus compounds from sewage treated in household sewage treatment plants. This solution is designed for facilities located within agglomerations and therefore requires an outflow with a phosphorus concentration below 2 or 1 mg P/L. For this reason, in MATERIALS AND METHODS we used similar solutions. The difference lies in the preparation of wastewater with a different composition. In these studies, where the wastewater contained only phosphorus compounds, we used tap water, which was associated with low COD and the presence of zinc compounds resulting from the material that is used in Poland to build water supply networks and internal water supply systems.
RESULTS AND DISCUSSION - are satisfactory; the description of figures is clear and understandable.
Thank you.
CONCLUSIONS - are general. The innovativeness of the conducted research was not emphasized.
We have corrected and supplemented the CONCLUSIONS.
REFERENCES - are adequate, correct and in line with the previous chapters. The use of the bibliography is correct.
Thank you.
Recommendation
I recommend this manuscript for publication in the journal Water with minor corrections.
Thank you.
Dear Reviewer 3
Thank You very much for Your comments.
Reviewer 4 Report
This study presents an important contribution to the field of wastewater treatment by investigating the effects of using light weight aggregates made of fly ash from sewage sludge thermal treatment (FASSTT LWA) on the effectiveness of phosphorus removal in vertical constructed wetlands (CW).
However, while the study provides valuable insights into the potential use of FASSTT LWA in improving phosphorus removal in CWs, there are some limitations that should be addressed.
For instance, the study was conducted over a relatively short period of 13 weeks, and it would be useful to investigate the long-term effects of FASSTT LWA in CWs.
Additionally, the study was limited to a single type of wastewater and did not evaluate the impact of FASSTT LWA on other pollutants that may be present in wastewater.
- Introduction should be rewritten. It should be expanded to include a more detailed discussion of current problems, and research gaps, especially the basic characteristics of the methods that authors is going to use, e.g., what’s the current methods to study FASSTT LWA, and why, what’s the advantage and disadvantages compare to other materials.
- It’s very strange that many single sentences are a paragraph.
- Novelty of the study, the authors need to point out their objectives clearly and novelty of the study, which need to be stressed in the revised version.
- The data processing and statistic analysis are missing, did the authors make any replicates for the experiment? If not, then the obtained results are doubtful.
Author Response
Dear Reviewer 4,
Thank You very much for Your comments.
These are our answers:
For instance, the study was conducted over a relatively short period of 13 weeks, and it would be useful to investigate the long-term effects of FASSTT LWA in CWs.
The research has lasted for 13 weeks, but before that, wastewater had been fed to VFCW for 30 days prior to the actual investigation and analytical control of the processes. It would definitely be useful to explore the long-term effects of LWA's FASSTT on CW. Unfortunately, for reasons beyond our control, the experiment could not last longer than 130 days. At the same time, we would like to point out that the concentration of phosphorus in the treated sewage was at a fairly stable level, which proves the potentially high "P absorption" of the filling.
Additionally, the study was limited to a single type of wastewater and did not evaluate the impact of FASSTT LWA on other pollutants that may be present in wastewater.
Wastewater containing phosphorus compounds with a concentration of about 7.43 mgPPO4/L corresponding to the amount of phosphorus found in wastewater flowing from household sewage treatment plants, was to be the subject of the research.
Low COD values and some amounts of nitrates could be expected in such wastewater. The synthetic wastewater COD was low, characteristic of tap water used for wastewater preparation, as was the concentration of Zn (galvanized steel pipes are used in Poland in water supply system and in plumbing), which must meet the values for drinking water. There were no nitrates in VFCW inflow, because the aim of the research was to determine the effect of changing the frequency of wastewater supply on improving the efficiency of phosphorus removal in VFCW with FASSTT LWA filling. Therefore, we focused exclusively on phosphorus compounds. But our previous research showed that FASSTT LWA can be used for nitrogen ammonia and nitrogen removal.
- Introduction should be rewritten. It should be expanded to include a more detailed discussion of current problems, and research gaps, especially the basic characteristics of the methods that authors is going to use, e.g., what’s the current methods to study FASSTT LWA, and why, what’s the advantage and disadvantages compare to other materials.
Introduction has been rewritten. We had shortened some paragraphs and added information about technological and environmental advantages of FASSTT LWA comparing natural materials. We emphasized the role of VFCW with FASSTT LWA in the treatment of wastewater discharged into the environment from household wastewater treatment plants located in the agglomeration through additional removal of phosphorus.
- It’s very strange that many single sentences are a paragraph.
We have improved it.
- Novelty of the study, the authors need to point out their objectives clearly and novelty of the study, which need to be stressed in the revised version.
We have completed this part of the article.
- The data processing and statistic analysis are missing, did the authors make any replicates for the experiment? If not, then the obtained results are doubtful.
“Each filling variant was repeated three times” – this information can be found in the summary (line 25) and in the subchapter entitled 2.1. Lysimeters (line 210). The obtained data were subjected to statistical analysis - descriptive statistics - the mean of three repetitions and the standard deviation were determined.
Dear Reviewer 4
Thank You very much for Your comments.
Round 2
Reviewer 2 Report
I read the revised version of manuscript Water-2315669 and, in my opinion, the problems which I highlighted in the first review remain. As for the previous version, my concerns regard:
The values of hydraulic loading rates. The authors say that an inadequate contact time between the filter medium and the wastewater do not allow the phosphorous adsorption, and this is reasonable, but the tested values (in the order of 5 mm/d) are applicable in real contexts?
The wastewater used for the test is not representative of a real wastewater, even after a biological treatment capable of removing most of the organic substance and phosphorous.
Most of the references dates back to before 2015 and the results are mainly discussed with reference to a previous study by the same authors (ref. n. 29 in Polish). For what concerns the other studies used to support the discussion, it is not clear if the conditions used to carry out those studies are similar to those of this study.
Reviewer 4 Report
The revised version has improved, I have no further comments.